# eRAM-V: From Interaction to Integration in Efficient Multimodal Large Language Models

## Abstract

Multimodal large language models (MLLMs) have made significant progress in recent years, yet the interaction between vision and language representations remains underexplored. Prior work has primarily relied on empirical heuristics to guide architecture design. While effective, this approach can lead to sub-optimal designs and computational redundancy. In this work, we examine the fusion process between visual and textual data. Our findings indicate that in auto-regressive MLLMs, fine-grained interactions between visual and text tokens primarily occur in the middle layers. This leads to redundancy in the shallow and deep layers, where modeling only selected visual representations is sufficient. Based on these insights, we introduce eRAM-V, an MLLM that balances computational efficiency and performance. eRAM-V models selected visual features across all layers and integrates fine-grained visual features at specific layers, as needed. Extensive experiments show that eRAM-V outperforms baseline models with equivalent computational budgets, achieving superior results across various benchmarks.

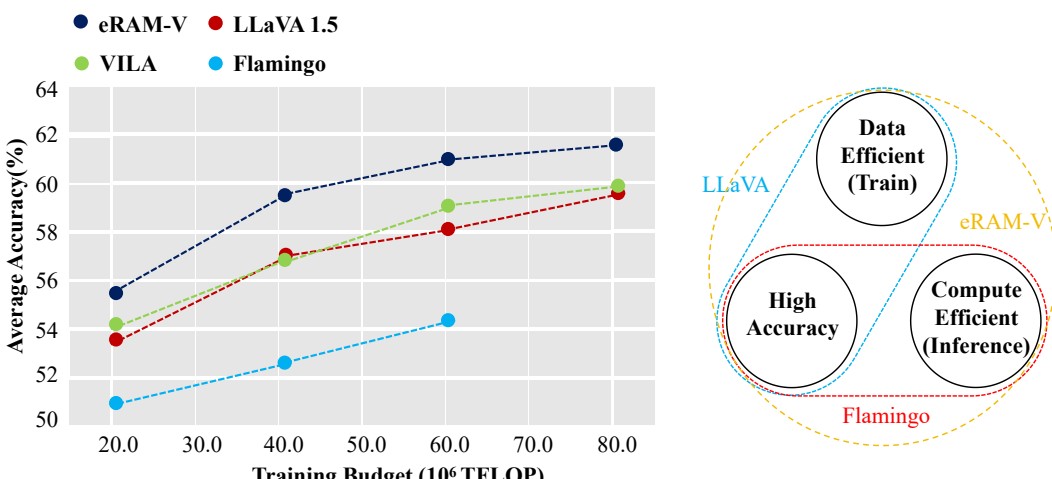

Figure 1: *Left*: Comparison of model performance under the same training budget. For ablation, we use the same dataset as eRAM-V to train LLaVA, VILA, and Flamingo. Although Flamingo operates with the smallest training budget, it struggles to perform on par with the others due to its reliance on a larger data scale. We report average accuracy across four benchmarks (TextVQA, GQA, MME, and Science-QA). *Right*: Auto-regressive models like LLaVA show strong performance and data efficiency in training, while the cross-attention-based Flamingo model offers faster inference but requires a notoriously large dataset for training. Our model strikes a balance between data efficiency, computational efficiency, and performance.

## 1 Introduction

Recent advancements in Multimodal Large Language Models (MLLMs) (Han et al., 2023; Wang et al., 2024; Han et al., 2024b; Liu et al., 2024c) have been driven by the successful integration

of pretrained vision encoders with LLMs, as seen in models like Flamingo (Alayrac et al., 2022), BLIP-2 (Li et al., 2023a), LLaVA (Liu et al., 2023), and MiniGPT-4 (Zhu et al., 2023), among others, marking a significant leap in the field. These MLLMs demonstrate a wide range of vision-language capabilities. For example, they can generate code from images, convert plots into Markdown tables, and even perform web browsing (Yang et al., 2023; Liu et al., 2024b; Hong et al., 2024). Despite their notable performance, the interactions between different modalities in MLLMs remain under-explored. Recent MLLM designs typically follow auto-regressive architectures (Lin et al., 2024), or incorporate modifications guided by empirical intuition (Wang et al., 2023). While these models achieve strong results, the lack of detailed investigation into modality interactions may result in unnecessary computational overhead, as we demonstrate in Section 3.

To address the limited exploration of modality interactions, recent approaches like FastV (Chen et al., 2024b) have focused on improving MLLM efficiency by refining vision-language interactions. FastV reduces up to 50% of visual tokens while maintaining overall performance. Another study leverages an "information flow" framework (Zhang et al., 2024) to analyze these interactions, revealing that most occur in the shallow layers. They propose token truncation in these layers based on attention scores, demonstrating improvements in reasoning capabilities. While these studies offer valuable insights, they lack comprehensive analyses. For instance, they do not evaluate performance across a broader range of perception tasks and overlook the exploration of more efficient learning modules that could better balance computational efficiency and performance.

In this paper, we present a comprehensive investigation of the vision-language interaction process, focusing on the analysis of cosine similarity between visual and text tokens, attention maps, and entropy (details in Section 3). Specifically, we study the impact of each input token on generating the final answer with given prompts. To achieve this, we visualize attention scores at each layer, tracking how attention scores contribute to the token generation. Our investigation of MLLMs focuses on two key perspectives: (1) the transformation of visual tokens, and (2) the interaction between visual and textual features. Through this analysis, we observe that redundancy in current MLLMs (*e.g.*, LLaVA) largely arises from modeling every fine-grained visual token across both the shallow and deep layers. Based on these insights, we propose eRAM-V[1], a model that balances data efficiency, computational efficiency, and performance (see Figure 1 for a comparative analysis).

The architecture of eRAM-V is designed with computational efficiency as its core principle. Rather than modeling all visual features as input embeddings, only selected tokens, identified through our analysis, are propagated throughout the entire MLLM. This significantly reduces the number of visual tokens and, consequently, the context length, aligning with our goal of optimizing computational efficiency. We define these selected tokens as *pseudo-global* tokens, since they originate from local regions but exhibit higher norms, capturing coarse global information, similar to the tokens observed in Vision Transformers (ViTs) (Darcet et al., 2023). As retaining only pseudo-global tokens may result in a loss of visual details, fine-grained visual tokens are selectively injected at critical layers, primarily in the mid-layers where interactions are most essential.

eRAM-V features a hierarchical design, employing different strategies for integrating visual information at various levels of granularity. Experimental results show that, under the same training budget, eRAM-V consistently outperforms LLaVA, VILA, and Flamingo in both inference efficiency and performance (see Figure 1 *Left*).

Our contributions can be summarized as follows:

- We conduct a comprehensive analysis of current multimodal large language models (MLLMs) by examining cosine similarity during vision transformations. We also analyzing token contributions via attention maps, and entropy in vision-language interactions. Our findings highlight significant computational redundancies in existing MLLMs (*e.g.*, LLaVA).

- Based on these insights, we propose eRAM-V, a novel MLLM architecture that models different levels of granularity, achieving a balance between data efficiency, computational efficiency, and performance.

---

[1]Why eRAM-V? Our model architecture **e**fficiently injects **V**isual features into MLLMs through an "highway **RAM**p", optimizing both computational and data efficiency.

- We perform extensive experiments and ablations across a diverse range of tasks to empirically validate the effectiveness and robustness of eRAM-V.

# 2 RELATED WORK

## 2.1 MULTIMODAL LARGE LANGUAGE MODELS

Multimodal Large Language Models (MLLMs) have significantly benefited from the advancements in Large Language Models (LLMs). MLLMs inherit the extensive world knowledge of LLMs while gaining cross-modal understanding through modality alignment. Notable examples include Flamingo (Alayrac et al., 2022), LLaVA (Liu et al., 2023; 2024e), Qwen-VL (Bai et al., 2023), and Idefics (Laurençon et al., 2024; Laurençon et al., 2024). These models can process images at various resolutions, from low to high, with increased accuracy and excel at complex reasoning tasks, such as image-conditioned code generation.

Althouth many MLLMs have been proposed, they generally fall into two structural categories. The first, *cross-attention-based* models, includes examples like Flamingo, Idefics1 (Laurençon et al., 2024), InfiMM-HD (Liu et al., 2024b), and EVLM Chen et al. (2024a). These models integrate visual information into textual inputs via gated cross-attention between the decoder layers of the LLM, offering high computational efficiency during inference. However, the introduction of cross-attention layers adds a substantial number of learnable parameters and disrupts the LLM's alignment, requiring substantially more data for re-alignment.

The second type is based on *auto-regressive* models, with LLaVA being a representative example. These models project input visual features into soft embeddings via a simple MLP, which are then concatenated with textual embeddings before being fed into the LLM. In addition to the simple MLP vision-language projection, various connectors have been proposed (Li et al., 2023a; Cha et al., 2024; Jian et al., 2024; Lin et al., 2024). This approach is typically more data-efficient, achieving competitive performance even with smaller datasets. However, it introduces higher computational complexity, as computation scales quadratically with input sequence length.

## 2.2 EXPLORATION OF MODALITY INTERACTIONS

Current research on MLLMs primarily focuses on enhancing perceptual capabilities by increasing input resolution and creating more diverse instruction-tuning datasets (Li et al., 2024; Chen et al., 2023b; Liu et al., 2023; Han et al., 2024a). However, considerably less attention has been given to understanding the contributions of individual tokens or features within these models, leaving many architectural decisions driven by empirical results rather than more systematic exploration. OPERA (Huang et al., 2024) is the first study to analyze the attention maps of MLLMs and identify potential causes of hallucinations. Their findings reveal that hallucinations during inference can occur when the model processes specific tokens, such as "-" and "?". To address this, they propose introducing penalty constraints on attention scores. Similarly, FastV (Chen et al., 2024b) examines attention mechanisms in MLLMs, highlighting inefficiencies in deeper layers where attention to visual tokens becomes sparse, leading to significant redundancy. To accelerate inference, they propose pruning visual tokens at specific layers, improving throughput while maintaining comparable performance. Another study (Zhang et al., 2024) introduces the concept of "information flow" to analyze interactions between visual and textual modalities during complex reasoning tasks. The authors also note that deeper layers suffer from redundancy and propose visual token pruning to help the model focus on the most important aspects of the input image. Their experimental results show that this approach enhances performance on reasoning tasks.

Although previous studies have investigate vision-language interactions in MLLMs to reduce computational costs, they have primarily focused on token pruning, a relatively simple approach that lacks deeper investigation into more complex strategies. In this work, we conduct a comprehensive analysis and, based on our findings, propose a novel architecture that provides balanced, holistic improvements in both efficiency and performance across a wide range of tasks.

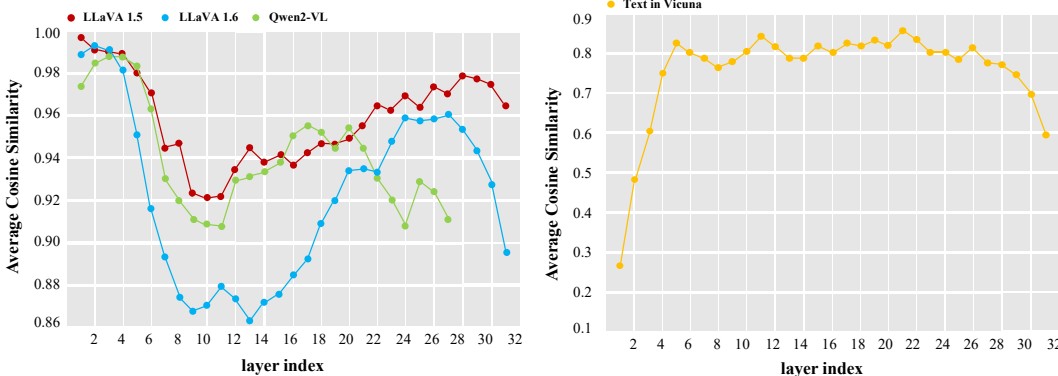

Figure 2: *Left*: Average cosine similarity (ACS) of visual features across layers in various MLLMs. The variation of visual features in MLLMs is minimal in both shallow and deep layers, as indicated by ACS scores approaching 1.0. In comparison, *Right*: ACS scores for text tokens in Vicuna typically range from 0.2 to 0.9.

# 3 METHODS

In this section, we begin with a comprehensive analysis of existing MLLMs, highlighting the key findings that inform our approach. Drawing from these insights, we present the eRAM-V architecture, which is specifically crafted to tackle the challenges identified in our analysis.

In most modern MLLMs, such as LLaVA and Qwen-VL, visual signals are transformed into visual tokens, which are then concatenated with textual inputs to form multimodal input sequences. As these inputs pass through the Large Language Model (LLM), attention mechanism can generally be divided into four categories: (1) text-to-text attention, (2) vision-to-vision interaction, (3) vision-to-text interaction, and (4) text-to-vision interaction. In this study, we focus on investigating the interaction between vision and language. To streamline the analysis, we restrict our scope to single-image scenarios, similar to the approach used in LLaVA-1.5. Specifically, our analysis focuses on two key areas: text-to-vision interaction and the multi-modal integration of visual and textual tokens. By concentrating on single-image scenarios, we streamline the investigation while preserving a strong focus on the interaction and integration between vision and language.

## 3.1 REDUNDANCY IN VISUAL TRANSFORMATION

We employ Average Cosine Similarity (ACS) as the primary monitoring metric to assess variations in visual tokens. Let the input visual tokens to the LLM be denoted as $\mathbf{V} \in \mathbb{R}^{N \times D}$, where $N$ represents the number of visual tokens, and $D$ denotes the feature dimension. At the $n$-th layer of the LLM, the input visual tokens are represented as $\mathbf{V_n} \in \mathbb{R}^{N \times D}$, and the corresponding output visual tokens are $\mathbf{V_{n+1}} \in \mathbb{R}^{N \times D}$. The ACS at layer $n$ is calculated as follows:

$$\text{ACS}_n = \frac{1}{N} \sum_{i=1}^{N} \frac{V_n^i \cdot V_{n+1}^i}{\|V_n^i\|_2 \cdot \|V_{n+1}^i\|_2}. \tag{1}$$

**Redundancy in Shallow Layers** In Figure 2, we show the ACS values across different layers for several representative MLLMs. The results indicate that the variation of visual tokens remains relatively minimal across both shallow and deep layers, with ACS values consistently near 1.0 for visual tokens, significantly higher than those for text tokens (About the text tokens' high yet relatively smaller cosine similarity than Visual tokens, in (Gromov et al., 2024), the authors highlight a notable similarity between text tokens within the LLM. Similarly, we observe a comparable phenomenon in pretrained ViT models, which we attribute to the residual connections inherent in transformers. These connections enable smoother transformations across layers. Despite this smoothness, e.g. roughly 0.8, the transformations still play a critical role in shaping the model's overall performance.), which range from 0.2 to 0.9. This finding motivates us to eliminate the costly transformations of visual token in these layers.

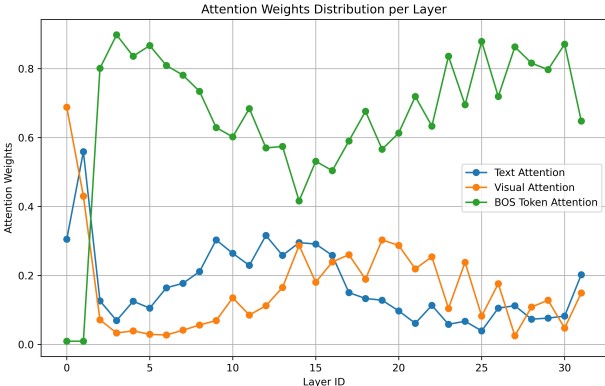

Figure 3: The attention distribution of generated tokens across layers. In shallow layers, the attention scores are concentrated on visual and text categories, whereas in deeper layers, the focus shifts toward special tokens.

**Redundancy in Middle Layer Self-Attention**  However, in the middle layers (8-24), the variation becomes more pronounced. To further investigate, we conduct an ablation study with LLaVA 1.5 where the visual tokens are selectively passed through either the Self-Attention or the Feed-Forward Network (FFN) in LLM, while text tokens continue to pass through both components. This selective approach allows us to isolate the contribution of each component for visual tokens. As shown in Appendix C, the FFN is primarily responsible for the observed changes, underscoring its critical role in model performance.

These findings highlight redundancy in both the shallow and middle layers, which inspire our design adjustments. In Section 3.3, we build on these insights by removing visual token processing in the shallow layers and refining self-attention in the middle layers, aiming to enhance computational efficiency without sacrificing performance.

## 3.2 VISION-TEXT INTERACTION

**Strong Attention to Visual Tokens in Shallow Layers**  To analyze the vision-text interaction process, we primarily examine the attention scores of each layer in the LLM decoder. In the case of LLaVA, the input embeddings of the LLM are grouped into three categories: visual tokens, text tokens, and special tokens (*e.g.*, begin_of_sentence). We investigate the attention score across different token categories for each layer. We sum the attention scores of the newly generated token to the prefix tokens from each of the three categories separately, as shown in Figure 3. The result indicates that attention score distribution concentrates on visual and text categories in shallow layers, while shifting focus to special tokens in deeper layers. This suggests that visual tokens hold greater significance in the shallow layers, consistent with the observations made in previous study (Zhang et al., 2024).

**Sparse and Consistent Attention to Visual Tokens in Shallow Layers**  To better understand such attention pattern, we visualize the attention map of generated tokens for visual tokens using a single example, as shown in Figure 4. The visualization indicates that, despite varying questions, the shallow layers of the LLM exhibit similar visual attention pattern. Specifically, attention is concentrated on a limited number of visual tokens, highlighted in the attention maps. To quantify this, we calculate the Jaccard distance of the distributions of the top 25 tokens across different prompts, resulting in a score of 0.7, which further aligns with our observations. Notably, these findings align with the visual anchors identified by Liu et al. (2024a). In the middle layers, attention shifts to specific image regions according to text prompts. For instance, when asked about the dog, layer 10 highlights the relevant area. When the question shifts to an object near the dog, the focus moves to the bike accordingly.

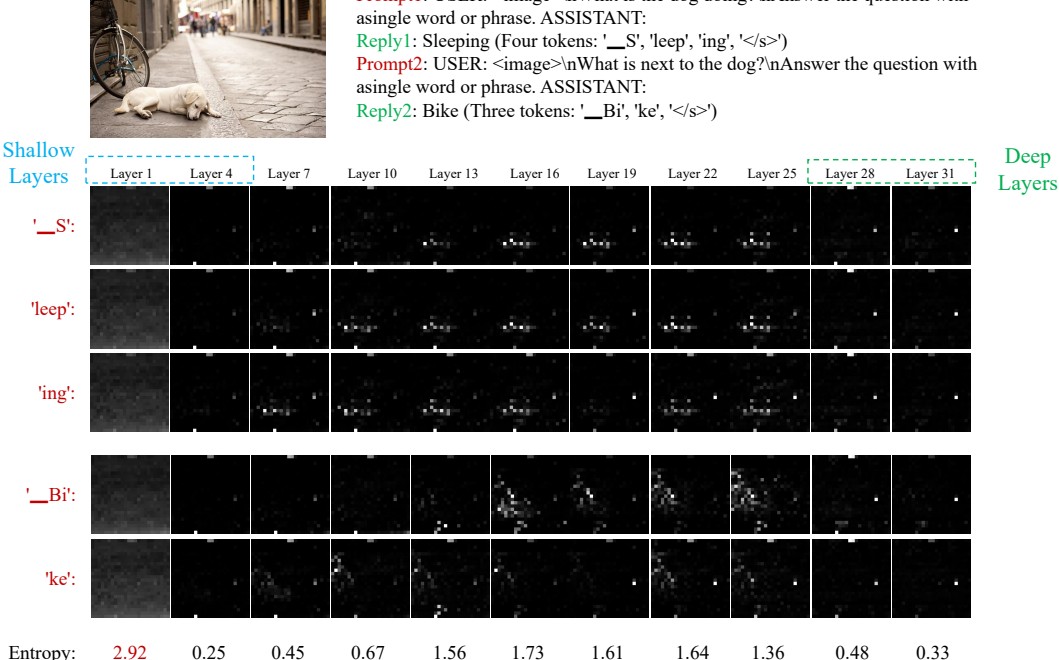

Figure 4: Examples of the attention map on visual tokens for each generated text token. Two different questions are asked based on the same image, with the model generating the outputs "sleeping" and "bike", respectively. The corresponding attention maps for the tokens "sleeping" and "bike" are shown in the upper three rows and the bottom two rows. Despite the different questions, the shallow layers of the LLM display similar visual attention patterns. In the middle layers, attention shifts to specific image regions based on the different questions. We also calculate the entropy of the attention patterns. In the shallow and deep layers (excluding the first layer), the entropy is relatively low, indicating simpler attention distributions. In contrast, the entropy is significantly higher in the middle layers, reflecting more complex attention patterns.

We also compute the entropy of the attention maps for each layer. Let the attention for the generated token in layer $n$ be $\mathbf{A_n} \in \mathbb{R}^{1 \times N}$, we calculate the entropy as:

$$\text{Entropy}_n = -\sum_{i=1}^{N} \mathbf{A_{ni}} \log \mathbf{A_{ni}}. \tag{2}$$

In the shallow and deep layers (excluding the first layer, where the attention pattern is deemed ineffective), the entropy is relatively low, indicating a simpler attention pattern. While in the middle layers, the entropy is large, align with the complex attention pattern. This is also aligned with the observation in Figure 2 that visual tokens change most in middle layers.

Based on these observations, we conclude that text to visual attention focus on general pattern of the image in shallow layers, while shifting to fine-grained patterns in the middle layers. Only a small subset of visual tokens is utilized across all decoder layers.

## 3.3 MULTIMODAL INTEGRATION

Based on the observations outlined above, we propose eRAM-V. From Section 3.1 and Section 3.2, we derive the following conclusions: (1) Only a small subset of visual tokens is engaged across all decoder layers (from shallow to deep), and these tokens demonstrate limited relevance to the provided instruction. (2) Fine-grained vision-text interactions are predominantly concentrated within the middle layers. (3) Retaining Feed-Forward computations for fine-grained visual features in the middle layers of the MLLM yields significant benefits, whereas such transformations are less critical in the shallow and deep layers. Given conclusion (1), we opt to retain a small portion of visual tokens

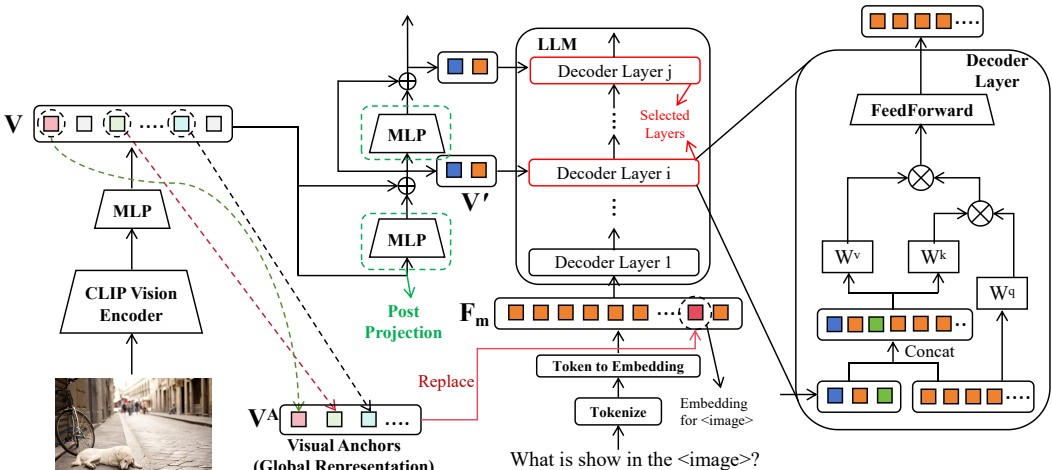

Figure 5: Architecture of eRAM-V.

as LLMs' input, to reduce computation redundancy. We extract the visual anchors as the pseudo-global feature representations, which are then concatenated with text tokens and fed into LLMs. In line with conclusion (2), we insert cross-attention between every two LLM decoder layers, from layers 8 to 22 (*i.e.*, 8, 10, 12, *etc.*). Unlike Flamingo, we reuse the LLM decoder layers' self-attention weights for cross-attention avoiding the need for extra parameters. This parameter reuse not only reduces model parameters, but also improves model convergence, requiring less training data. Finally, based on conclusion (3), we introduce Post-Projection layers to facilitate further adjustments to the visual tokens. By building on the strengths of both Flamingo and LLaVA, we develop eRAM-V.

The architecture of eRAM-V is depicted in Figure 5. Given an input image $\mathbf{I} \in \mathbb{R}^{C \times H \times W}$ and a visual encoder $F_v$, where $C$, $H$, and $W$ represent the image channels, height, and width respectively, we can obtain the visual features $\mathbf{V} \in \mathbb{R}^{N \times D}$ and the corresponding attention map of the [CLS] token $\mathbf{A} \in \mathbb{R}^{h \times 1 \times N}$ using the following equation:

$$\mathbf{V}, \mathbf{A} = F_v(\mathbf{I}), \tag{3}$$

where $N$, $D$, and $h$ denote the total number of visual tokens, the dimension of the visual tokens, and the number of attention heads, respectively. To identify the visual anchors, we first rank the visual tokens based on their associated attention weights. The top-$k$ visual tokens are selected and then sorted according to their original index (in ascending order). This process yields the visual anchors $\mathbf{V_A} \in \mathbb{R}^{M \times D}$, where $M$ denotes the number of visual anchors. Next, a shared MLP module is used to project both $\mathbf{V}$ and $\mathbf{V_A}$ to the dimensionality required by the LLM, resulting in $\mathbf{V}'$ and $\mathbf{V'_A}$. Following LLaVA's approach, the visual anchors $\mathbf{V'_A}$ are concatenated with the text embeddings to form the multimodal input for the LLM. The detailed visual features $\mathbf{V}'$ are used within the LLM's decoder layers to enhance fine-grained understanding. For example, in decoder layers $[8, 10, 12, 14, 16, 18, 20, 22]$, these fine-grained features provide more detailed visual information.

Let the multimodal input embeddings be denoted as $\mathbf{F_m} \in \mathbb{R}^{(M+T) \times D}$, where $T$ represents the number of text tokens. For decoder layers that do not leverage fine-grained visual information, the computation proceeds as follows:

$$\mathbf{O_1} = \mathbf{F_m} + \text{Attn}(q = \mathbf{F_m}, k = \mathbf{F_m}, v = \mathbf{F_m}), \tag{4}$$

$$\mathbf{O_2} = \mathbf{O_1} + \text{FF}(\mathbf{O_1}), \tag{5}$$

where Attn refers to the causal attention mechanism within the LLM decoder layer, and FF represents the Feed-Forward module within the same layer.

In contrast, for decoder layers utilizing fine-grained visual features, the computation is as follows:

$$\mathbf{K} = \text{Concat}(\mathbf{V}', \mathbf{F_m}), \tag{6}$$

$$\mathbf{V} = \text{Concat}(\mathbf{V}', \mathbf{F_m}), \tag{7}$$

$$\mathbf{O_1} = \mathbf{F_m} + \text{Attn}(q = \mathbf{F_m}, k = \mathbf{K}, v = \mathbf{V}). \tag{8}$$

In this framework, no additional cross-attention layers are introduced, maintaining data efficiency without significantly increasing the number of parameters. Furthermore, by concatenating the visual anchors directly into the multimodal input embeddings, we significantly reduce the computational overhead. Consequently, this design allows our eRAM-V architecture to achieve a balanced integration of data efficiency and computational cost.

## 4 EXPERIMENTS

### 4.1 SETTINGS

**Benchmarks** We employ a diverse set of seven benchmarks to thoroughly evaluate the overall performance of our proposed model. Our evaluation primarily emphasizes two key aspects: reasoning ability and perception ability. For instance, Science-QA, which includes questions related to scientific knowledge, measures the model's capacity for complex reasoning. In contrast, benchmarks such as TextVQA assess the model's ability to perform fine-grained visual perception tasks. A detailed overview of these benchmarks is provided in Appendix A.

**Implementation Details** In our main experiments, we use the Vicuna-v1.5 model as the LLM (Chiang et al., 2023) (Ablations on different LLMs are shown in Appendix D). Following LLaVA-1.5, CLIP ViT-L/14, pre-trained at $336 \times 336$ resolution is used as the visual encoder, and we use the penultimate layer's output as the vision feature. Our training dataset is based on LLaVA-1.5 (Liu et al., 2024d), consisting of 558k samples for pre-training and 665k samples for instruction tuning. We also incorporate a multi-task training phase with data sampled from various QA benchmarks (details on training data are provided in Appendix B). We adopt a three-stage training framework (see Appendix H for visualizations). In the pretraining stage, only the MLP layers are trainable. In the subsequent continued pretraining phase, both the vision encoder and MLP layers are unfrozen. Finally, during the instruction fine-tuning stage, the vision encoder is frozen while the remaining modules are trainable. Despite the additional training stages, our approach results in lower computational costs compared to LLaVA for both training and inference.

### 4.2 MAIN RESULTS

The main results are presented in Table 1 and Table 2. Our model exhibits strong performance, achieving results comparable to the original LLaVA-1.5 model while significantly reducing inference FLOPs, despite being trained on a limited dataset. This performance is consistent across various benchmarks, including those requiring advanced visual perception (*e.g.*, TextVQA) and overall capability (*e.g.*, MME). However, our model shows slightly lower performance on specific benchmarks like GQA compared with LLaVA-1.5. Given the substantial reduction in computational costs, this minor decline is expected, and the performance gap remains small and negligible relative to the significant improvements in efficiency and speed. Additionally, we validate the effectiveness of eRAM-V using other LLMs, with the results presented in Appendix D).

Additionally, we compare the performance of four architectures, eRAM-V, VILA, Flamingo, and LLaVA, under varying training budgets (measured in TFLOPs). In this comparison, all models utilize ViT-L as the vision encoder and Vicuna-7b as the LLM. We report the average accuracy across four benchmarks: TextVQA, GQA, MME, and Science-QA. eRAM-V's redundancy reduction strategy enables it to achieve significantly better performance under the same training budgets (as shown in Figure 1 *left*).

### 4.3 ABLATION STUDIES

In this module, we mainly analyse the performance on accuracy. We also add ablation study on efficiency in Appendix F.

Table 1: Evaluation results for general MLLM benchmarks show that eRAM-V achieves equal or superior performance compared to other models, while significantly reducing computational costs (measured in TFLOPs).

| Model | LLM | Res | POPE | MME$^p$ | MME$^c$ | SQA$_{img}$ | TFLOPs ($\downarrow$) |
|---|---|---|---|---|---|---|---|
| *Approaches using 7B Large Language Models* | | | | | | | |
| InstructBLIP(Dai et al., 2023) | LLaMA2-7B | 224 | 78.9 | - | - | 60.5 | - |
| Qwen-VL-Chat(Bai et al., 2023) | Qwen-7B | 448 | 85.9 | 1487.5 | - | 68.2 | - |
| LLaVA-1.5 (Liu et al., 2024d) | Vicuna-7B | 336 | 85.9 | **1510.7** | 260.3 | 66.8 | 13.32 |
| eRAM-V | Vicuna-7B | 336 | **86.8** | 1490.8 | **315.3** | **69.1** | **4.43** |
| *Approaches using 13B Large Language Models* | | | | | | | |
| InstructBLIP(Dai et al., 2023) | Vicuna-13B | 224 | 78.9 | 1504.6 | - | 63.1 | - |
| BLIP-2(Li et al., 2023a) | Vicuna-13B | 224 | 85.3 | 1293.8 | 290.0 | 61.0 | - |
| LLaVA-1.5 (Liu et al., 2024d) | Vicuna-13B | 336 | 85.9 | 1531.3 | 295.1 | 71.6 | 25.14 |
| eRAM-V | Vicuna-13B | 336 | **86.8** | **1533.6** | **330.3** | 71.6 | **8.17** |

Table 2: Results on VQA benchmarks. eRAM-V demonstrates strong performance on TextVQA and slightly underperforms on GQA, while significantly reducing computational resources. [†]: Image-only input.

| Model | LLM | Res | TextVQA[†] | GQA | OKVQA |
|---|---|---|---|---|---|
| *Approaches using 7B Large Language Models* | | | | | |
| InstructBLIP(Dai et al., 2023) | Vicuna-7B | 224 | 50.7 | 49.2 | - |
| Shikra (Chen et al., 2023a) | Vicuna-7B | 224 | - | - | 47.2 |
| IDEFICS-9B (Laurençon et al., 2024) | LLaMA-7B | 224 | - | 38.4 | 49.6 |
| BLIP-2(Li et al., 2023a) | Vicuna-7B | 224 | 40.1 | 38.6 | - |
| Qwen-VL(Bai et al., 2023) | Qwen-7B | 448 | - | 59.3 | 58.6 |
| Qwen-VL-Chat(Bai et al., 2023) | Qwen-7B | 448 | - | 57.5 | 56.6 |
| LLaVA-1.5 (Liu et al., 2024d) | Vicuna-7B | 336 | 47.1 | **62.0** | 57.1 |
| eRAM-V | Vicuna-7B | 336 | **52.1** | 61.3 | **58.7** |
| *Approaches using 13B Large Language Models* | | | | | |
| InstructBLIP(Dai et al., 2023) | Vicuna-13B | 224 | - | 33.4 | - |
| BLIP-2(Li et al., 2023a) | Vicuna-13B | 224 | 42.5 | 41.0 | 59.3 |
| LLaVA-1.5 (Liu et al., 2024d) | Vicuna-13B | 336 | 50.2 | **63.3** | 60.0 |
| eRAM-V | Vicuna-13B | 336 | **52.6** | 62.0 | **60.4** |

**Ablations on Layers with Visual Injection**   In Flamingo, cross-attention layers are typically inserted uniformly across the model's decoder layers. However, our findings suggest that most fine-grained vision-text interactions occur in the middle layers of MLLMs. In this ablation study, we compare our default eRAM-V configuration, which inserts cross-attention in the middle layers (Layer ID: [8, 10, .., 20, 22], as shown in row 4 of Table 3), against uniform insertion across the model (Layer ID: [4, 8, .., 28, 32], as shown in row 1 of Table 3). The results show that incorporating fine-grained information in the middle layers leads to improved performance.

**Ablations on Visual Anchors and Fine-Grained Visual Information**   We present a detailed comparison of the impact of retaining or removing visual anchors from the input, as well as the performance differences between using only visual anchors and integrating them with fine-grained visual information. The results, shown by comparing row 2 and row 4 in Table 3, indicates that including visual anchors significantly boosts the model's performance in both cognitive and perceptual tasks. Moreover, when fine-grained visual information is combined with visual anchors, the model demonstrates notable improvement in perception, especially in tasks that require more detailed visual comprehension, as evidenced by the comparison between row 3 and row 4 in Table 3.

These findings highlight the critical role of visual anchors in providing a strong foundation for cognitive tasks, while the addition of fine-grained information further refines and enhances the model's perceptual capabilities. The combination of both visual anchors and fine-grained visual details is therefore essential for maximizing the model's performance across a diverse range of benchmarks. This comprehensive analysis highlights the necessity of preserving both types of visual information to achieve stronger results in multimodal large language models.

Table 3: "With AR" refers to the approach where visual anchors are selected and concatenated with the text embeddings to create the multimodal input embedding. "With CA" denotes the use of fine-grained visual information as additional key and value inputs in specific layers. "Layer Id" indicates the index of the layers where this occurs. When "With CA" is ✗, it indicates that no fine-grained visual information is used, and therefore "Layer Id" is marked as "-".

| Model | With AR | With CA | Layer Id | TextVQA | MME$^p$ | POPE | SQA$_{img}$ |
|---|---|---|---|---|---|---|---|
| eRAM-V | ✓ | ✓ | [4, 8, .., 28, 32] | 50.7 | 1460.4 | 86.3 | 67.4 |
| eRAM-V | ✗ | ✓ | [8, 10,.., 20, 22] | 44.8 | 1378.2 | 85.0 | 67.2 |
| eRAM-V | ✓ | ✗ | - | 50.0 | 1438.0 | 84.7 | 66.5 |
| eRAM-V | ✓ | ✓ | [8, 10,.., 20, 22] | **52.1** | **1490.8** | **86.8** | **69.1** |

**Ablations on ViT Post-Projection** In eRAM-V, we apply post-projection layers to the output of the ViT (two vertical MLP modules shown in Figure 5). In this ablation, we compare our default design with a variant that excludes the ViT post-projection. The results, shown in Table 4, demonstrate that this module consistently enhances the MLLM's performance across the given tasks.

Table 4: Ablation on ViT post-projection. eRAM-V with ViT post-projection consistently outperforms the variant without this module.

| Model | With Post-Proj | TextVQA | MME$^p$ | POPE | SQA$_{img}$ |
|---|---|---|---|---|---|
| eRAM-V | ✗ | 50.3 | 1431.6 | 86.2 | 64.6 |
| eRAM-V | ✓ | **52.1** | **1490.8** | **86.8** | **69.1** |

## 5 CONCLUSIONS

In this paper, we present a comprehensive analysis of common MLLMs, identifying architectural redundancies. Our findings lead to two key conclusions: (1) In both shallow and deep layers, the transformation of visual tokens within the LLM is largely redundant, with the Feed-Forward transformations in the middle layers being sufficient to capture the necessary information; (2) the integration of fine-grained vision and text features predominantly takes place in the middle layers of MLLMs. Building on these insights, we introduce eRAM-V, a novel MLLM architecture that achieves a balance between performance, computational cost, and training data efficiency. eRAM-V reduces redundancy through the use of visual anchors and sparse fine-grained vision-text integration. Extensive experiments show that, with the same training budget, our approach outperforms VILA, Flamingo, and LLaVA, while also offering lower inference latency compared to LLaVA.

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

# A    DETAILS ON CHOSEN BENCHMARKS

Below, we provide descriptions of tasks and metrics of the benchmarks used in this study. The chosen benchmarks are designed to evaluate both the general world knowledge of our models and their perceptual capabilities.

Table 5: Details on the chosen benchmark.

| Benchmark | Description of the task | Metric |
|---|---|---|
| OKVQA (Marino et al., 2019) | QAs about prior knowledge | VQA score ($\uparrow$) |
| TextVQA (Singh et al., 2019) | QAs about text in image (Visual Perception) | VQA score ($\uparrow$) |
| GQA (Hudson & Manning, 2019) | QAs of real world comprehension and complex reasoning | EM ($\uparrow$) |
| POPE (Li et al., 2023b) | QAs for Object Hallucination evaluation | F1 Score ($\uparrow$) |
| Sci-QA(Img) (Lu et al., 2022) | QAs about Science | Accuracy ($\uparrow$) |
| MME (Fu et al., 2023) | Comprehensive Evaluation Benchmark for MLLMs | Accuracy ($\uparrow$) |

# B    DETAILS ON TRAINING DATA

We present the details of the continue pretrain dataset below. All datasets are sourced from publicly available collections, reformatted to fit the instruction fine-tuning paradigm. In total, there are approximately 707k samples, each representing a single round of question-and-answer interaction. We mainly choose the dataset to enhance the visual perception ability as we finetuning Vision Transformer at this stage.

Table 6: Details on the training data of Continue Pretrain.

| Task | Dataset | Samples |
|---|---|---|
| General VQA | VQAv2 (Antol et al., 2015) | 221k |
| | OKVQA (Marino et al., 2019) | 9k |
| | VizWiz VQA (Gurari et al., 2018) | 20k |
| | GQA (Hudson & Manning, 2019) | 235k |
| Text-oriented VQA | TextVQA (Singh et al., 2019) | 34k |
| | OCRVQA (Mishra et al., 2019) | 83k |
| | DocVQA (Mathew et al., 2021) | 63k |
| | ChartQA (Masry et al., 2022) | 42k |
| Total | - | 707k |

# C    ABLATION STUDIES ON THE SELF-ATTENTION MECHANISM IN MLLMs

Table 7 illustrates the effects of separately removing the self-attention and feedforward computations for visual tokens in MLLMs. The results indicate that the feedforward computation is more critical than the self-attention mechanism for visual tokens, suggesting a degree of potential redundancy.

Table 7: Ablation studies on self-attention and feed-forward layers in LLaVA: We conduct these experiments using **Openllama-3b** as the LLM. "With SA" indicates retaining self-attention computation for visual tokens in the LLM, while "With FF" refers to retaining feed-forward computation for visual tokens.

| Model | With SA | With FF | TextVQA | MME$^p$ | POPE | SQA$_{img}$ |
|---|---|---|---|---|---|---|
| LLaVA 1.5 | ✗ | ✓ | 45.98 | 1420.3 | 86.6 | 59.3 |
| LLaVA 1.5 | ✓ | ✗ | 44.31 | 1398.2 | 85.4 | 59.1 |
| LLaVA 1.5 | ✓ | ✓ | 46.15 | 1440.4 | 86.7 | 59.1 |

# D  ABLATION STUDIES ON LLMs WITHIN MLLMs

To evaluate the robustness of eRAM-V, we conduct experiments using a variety of LLMs. Specifically, we test eRAM-V with both LLaMA3-8b and Mistral-7b to assess its adaptability across different architectures. The results, shown in Table 8, demonstrate that eRAM-V consistently delivers comparable performance across both models. This suggests that the model's efficiency and effectiveness are not dependent on a specific LLM but are broadly applicable.

eRAM-V's ability to maintain strong performance across different configurations highlights its versatility and robustness.

Table 8: Ablation studies on various LLMs are conducted, with all models utilizing the CLIP-ViT-L-336 as the Vision Encoder. RAM-V demonstrates comparable performance across all three models.

| Model | LLM | TextVQA | OKVQA | GQA | MME$^p$ | MME$^c$ | POPE | SQA$_{img}$ |
|---|---|---|---|---|---|---|---|---|
| eRAM-V | Vicuna-7b | 52.1 | 58.7 | 61.3 | 1490.8 | 315.3 | 86.8 | 69.1 |
| eRAM-V | LLaMA3-8b | 49.4 | 58.7 | 62.0 | 1533.6 | 330.3 | 86.8 | 69.3 |
| eRAM-V | Mistral-7b | 48.9 | 56.4 | 61.6 | 1406.5 | 320.1 | 86.4 | 69.1 |

# E  ADDITIONAL QUALITATIVE VISUALIZATIONS

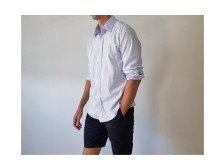

Prompt1: USER: <image>\nWhat is the man wearing?\nAnswer the question with a single word or phrase. ASSISTANT:
Reply1: Shirt (Three tokens: '__Sh', 'irt', '')
Prompt2: USER: <image>\nWhat is the color of the T-Shirt?\nAnswer the question with asingle word or phrase. ASSISTANT:
Reply2: White (Two tokens: '__White', '')

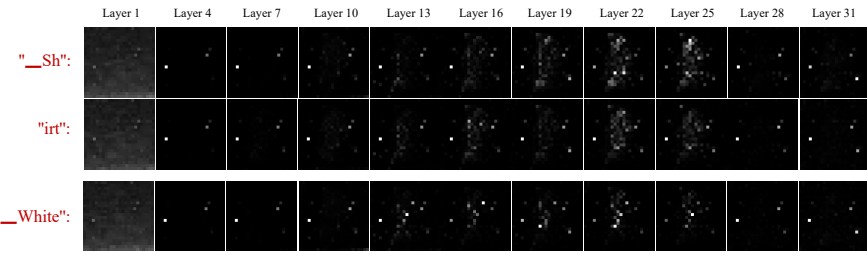

Figure 6: Additional qualitative visualizations of the attention maps during token generation. Two different questions (Prompt1 and Prompt2) are asked based on the same image, with the model generating the outputs "shirt" and "white", respectively. The corresponding attention maps for the tokens "shirt" and "white" are shown in the upper 2 rows and the bottom 1 rows. Despite the different questions, the shallow layers of the LLM display similar visual attention patterns. In the middle layers, attention shifts to specific image regions based on the different questions.

In this section, we provide additional qualitative visualizations of the attention maps during token generation. This analysis is based on LLaVA-1.5, evaluated on text-oriented tasks. As shown in Figures 6 and 7, fine-grained vision-text interactions predominantly occur in the middle layers. Specifically, for different prompts, the attention in the mid-layers focuses on distinct regions relevant to the given prompts.

# F  EFFICIENCY ANALYSE

In this module, we add more analyses on the efficiency. We first add FLOPs for other MLLM like BLIP2 or IDEFICS-9B. From Table 9 and Table 2, it can be found that our model achieves best balance considering the computing efficiency and accuracy. We also add the ablation on CA and AR here in Table 10 to individually test their influence on the final computation cost.

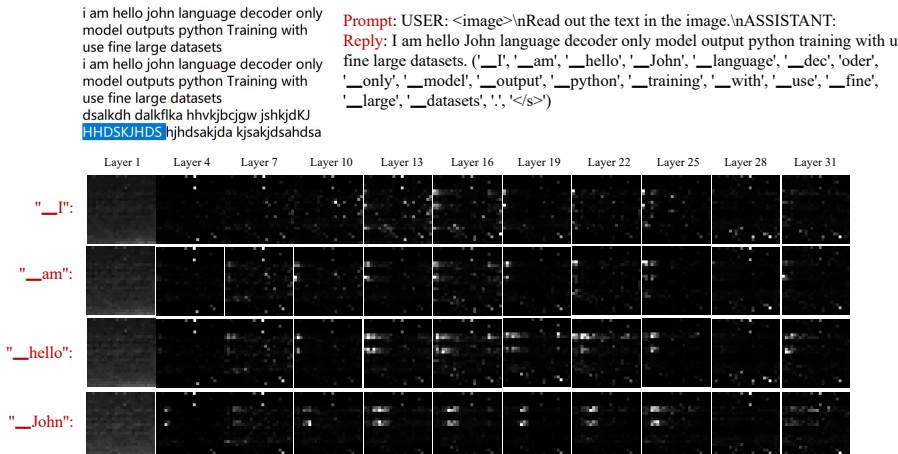

Figure 7: Additional qualitative visualizations of the attention maps during token generation.

Table 9: Flops for other MLLMs.

| Model | TFLOPs (↓) |
|---|---|
| BLIP2-7b | 3.12 |
| IDEFICS-9B | 2.78 |
| Qwen-VL-Chat 7b | 7.23 |
| LLaVA1.5 7b | 13.32 |
| eRAM-V 7b | 4.43 |

## G    RESULT FOR CAPTION

We add result for image caption here, which typically require richer visual information beyond just the main objects in the scene. Through Table 11, it can be found that our method also performs well in image caption task.

## H    VISUALIZATION OF OUR TRAINING STAGES

We adapt a three-stage training framework for eRAM-V, shown in Figure 8. During the pretraining stage, only the MLP layers are kept trainable. In the following continued pretraining phase, both the vision encoder and MLP layers are unfrozen. Lastly, during the instruction fine-tuning stage, the vision encoder is frozen, and the remaining modules are trained.

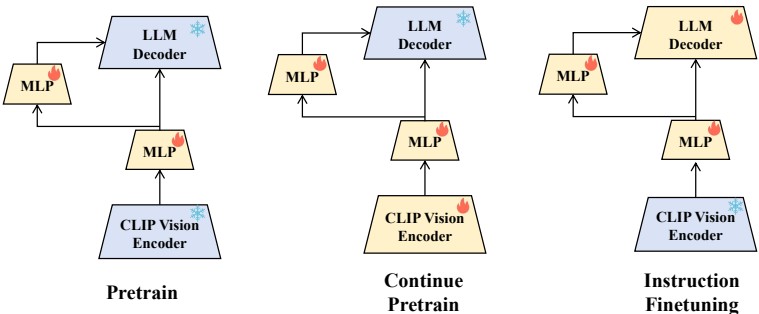

Figure 8: Visualization of our three-stage training framework.

Table 10: Flops with ablation on modules.

| Model | TFLOPs (↓) |
|---|---|
| eRAM-V 7b w/o CA | 3.54 |
| eRAM-V 7b w/o AR | 2.56 |
| eRAM-V 7b | 4.43 |

Table 11: Results for image caption task

| Model | NoCaps | Flickr30k |
|---|---|---|
| LLaVA1.5 7b | 99.8 | 67.9 |
| eRAM-V 7b | 100.1 | 68.2 |
| LLaVA 13b | 102.8 | 73.0 |
| eRAM-V 13b | 103.0 | 73.6 |

## I    NUMBER OF VISUAL TOKENS CHOSEN AS ANCHORS AND SELECTION CRITERIA

In this study, we select 65 visual anchors based on the weighted attention of the [CLS] token in the Vision Transformer (ViT). Specifically, visual tokens are ranked by their attention scores to the [CLS] token, where the attention values are averaged across the attention heads (reducing from dimensions $b \times h \times 1 \times n$ to $b \times 1 \times n$). The top 64 visual tokens, together with the [CLS] token, are then chosen to construct a final set of 65 visual anchors.

We explore three configurations of visual anchors in our experiments: 65, 145, and 257. These configurations yield comparable performance when fine-grained interaction is enabled through cross-attention. However, when the fine-grained interaction is removed, performance consistently improves as the number of visual anchors increases from 65 to 257. Despite this trend, we adopt the 65-anchor configuration as it provides a more efficient solution while leveraging the benefits of cross-attention.

## J    MORE VISUALIZATION ON VISION TRANSFORMATION PROCESS

We validate the transformation of visual tokens within LLMs is largely redundant in both shallow and deep layers with more dataset in Figure 9.

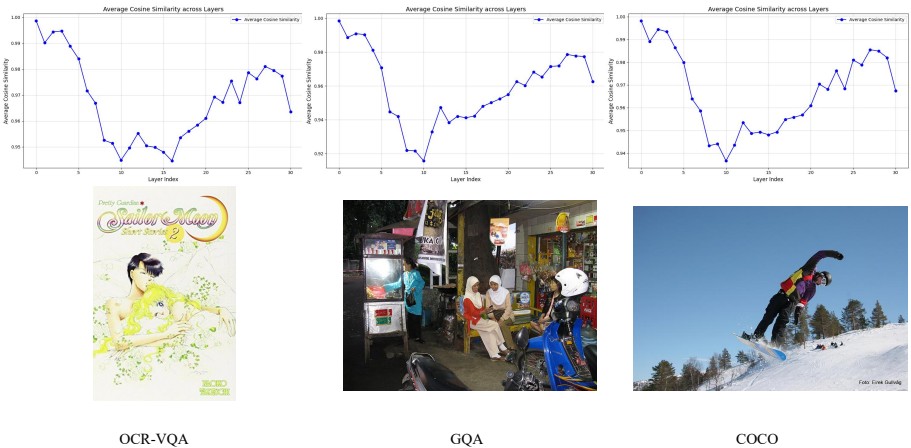

Figure 9: We show visual transformation process of some famous dataset.

