# OpenReview forum: "eRAM-V: From Interaction to Integration in Efficient Multimodal Large Language Models"
_ICLR.cc/2025/Conference — Submitted to ICLR 2025_

### Official Review · Reviewer_v1GT · 2024-10-29

**Soundness:** 2
**Presentation:** 3
**Contribution:** 3
**Rating:** 3
**Confidence:** 4

**Summary:**

This paper introduces eRAM-V, an efficient architecture for multimodal large language models (MLLMs) that optimizes the integration of visual and text data. Traditional MLLMs often rely on heuristic design choices, which may lead to suboptimal architectures and computational redundancy. eRAM-V addresses this by analyzing where in the network vision-text interactions occur most effectively, finding that fine-grained interactions primarily occur in the middle layers. Therefore, the model reduces redundancy by only retaining selected visual tokens in shallow and deep layers, while concentrating fine-grained vision-text interactions in the middle layers.

**Strengths:**

- Efficient and performance-oriented architecture design: eRAM-V effectively addresses visual redundancy by focusing fine-grained visual-text integration in the middle layers, reducing computational overhead while maintaining high-quality multimodal representation, which significantly improves inference efficiency.

- Comprehensive experimental validation and strong performance: eRAM-V demonstrates superior performance across various vision-language tasks, such as TextVQA and Science-QA, consistently outperforming existing models under the same computational budget, thus confirming the effectiveness of its design in real-world applications.

**Weaknesses:**

- Disconnect between proposed methods and theoretical analysis: Although the paper identifies high attention to visual tokens in the initial layers, it nonetheless chooses to prune tokens in these shallow layers, contradicting its own observations. This approach also seems inconsistent with the findings in FastV, which showed performance degradation when pruning visual tokens in early layers, raising questions about the rationale behind eRAM-V’s design choices.

- Logical issues in redundancy inference: The paper uses cosine similarity between visual and output tokens as an indicator of redundancy in visual tokens. However, it is unclear why similarity between visual tokens and outputs alone would imply redundancy; high similarity between visual tokens themselves would be a more convincing basis for this conclusion. This choice of metric could lead to questionable insights about redundancy.

- Limited experimental validation: The study only evaluates visual token pruning on VQA tasks, where core image content tends to suffice, minimizing impact from auxiliary details. This narrow evaluation leaves open questions about eRAM-V’s effectiveness on tasks like image captioning, which typically require richer visual information beyond just the main objects in the scene.

**Questions:**

1. Rationale for deducing redundancy based on cosine similarity between visual tokens and output tokens: The paper infers redundancy among visual tokens based solely on their cosine similarity with output tokens. However, wouldn’t redundancy be more appropriately established through high similarity among the visual tokens themselves, indicating overlapping information in their representations? Could you clarify why only visual-output token similarity was used to infer redundancy?

2. Contradiction in using FFN and cross-attention modules in deep layers: The paper states that in the deeper layers, the FFN module is primarily responsible for processing visual information, yet cross-attention modules are still added to capture extra image information. Doesn’t this design choice seem contradictory to the earlier analysis? Could you clarify whether cross-attention meaningfully contributes to visual processing in the deeper layers?

3. Importance and role of shallow-layer visual tokens: Although the paper suggests that self-attention in shallow layers has limited effect on visual information, attention distribution shows high attention to visual tokens in early layers (e.g., layers 1 and 2). Is the decision to emphasize fine-grained visual information in deeper layers based primarily on saliency analysis in middle layers? In contrast, FastV [1] shows that pruning visual tokens in shallow layers leads to a significant drop in performance, which appears to contradict this paper's conclusion on shallow-layer redundancy. Could you provide a more detailed explanation of the significance of shallow-layer visual information?

4. Contradiction with VTW [2] results regarding middle-layer importance: VTW shows that removing all visual tokens after layer 15 does not affect model performance, which seems inconsistent with this paper’s assertion of the importance of visual tokens in the middle layers. Could you discuss the differences between VTW and eRAM-V in terms of middle-layer visual token needs, and whether this suggests possible optimizations in eRAM-V’s middle-layer design?

5. Number of visual tokens chosen as anchors and selection criteria: How many visual tokens are ultimately selected as visual anchors, and what are the specific criteria for ranking these tokens based on CLS attention scores? Could you clarify how these parameters are determined and adjusted?

6. Clarification on the meaning of “text-to-image”: In autoregressive models like LLaVA and InstructBLIP, processing is typically image-to-text. Could you explain why the paper refers to text-to-image in line 184-185? Does the paper involve interleaved input of images and text? Additionally, was there a specific reason for placing the image after the question in the proposed framework?

7. MME dataset experimental results: Figure 1 includes the MME dataset in the experiments but seems to report average accuracy rather than the final MME score. What was the rationale behind this approach? Also, are the results shown in Table 1 obtained using the same data and training strategy?

If the author can address my questions, I will raise my rating.

[1] Chen, Liang, et al. "An image is worth 1/2 tokens after layer 2: Plug-and-play inference acceleration for large vision-language models." arXiv preprint arXiv:2403.06764 (2024).
[2] Lin, Zhihang, et al. "Boosting Multimodal Large Language Models with Visual Tokens Withdrawal for Rapid Inference." arXiv preprint arXiv:2405.05803 (2024).

---

### Official Review · Reviewer_HEaa · 2024-10-30

**Soundness:** 3
**Presentation:** 2
**Contribution:** 3
**Rating:** 6
**Confidence:** 4

**Summary:**

This paper introduces eRAM-V, a multimodal model optimized for vision-language fusion, focusing on integrating visual features in key layers to boost efficiency. eRAM-V achieves superior performance with less computational redundancy than existing models.

**Strengths:**

- The paper is well-written and easy to understand, making it accessible to a broad audience.
- The motivating ideas for the approach are both interesting and intuitive. The authors highlight the redundancy issue across layers, explaining that most interactions between visual and text tokens occur in the middle layers.
- The proposed method reduces the training budget while achieving comparable performance, showcasing an efficient approach to multimodal model training.

**Weaknesses:**

- Since one of the main contributions is proposing an effective method to reduce computational cost, the paper does not sufficiently discuss or experiment on this aspect.
- It would be better to analyze and compare the computational cost of the proposed method with other baselines in the Experiment Section.
- Table 1 lacks computational cost details for BLIP and QWen.
- There is no comparison of computational resources between each method in Table 2.
- In the ablation study of the AR and CA components, the paper should also report the computational cost of different settings to show how these components affect computational resource usage and performance.

**Questions:**

Please refer to the Weaknesses section for the list of questions.

---

### Official Review · Reviewer_M9GJ · 2024-11-02

**Soundness:** 4
**Presentation:** 3
**Contribution:** 3
**Rating:** 5
**Confidence:** 4

**Summary:**

The paper analyzes the redundancy of visual token inputs across different layers of multi-modal LLMs and proposes eRAM-V, a multi-modal integration framework that (a) replaces lengthy visual tokens in the multi-modal input sequence with shorter and more informative visual anchors, (b) introduces fine-grained visual-text token interactions through a parameter-shared cross-attention mechanism in the middle layers of the decoder LLM, and (c) retains direct visual features interaction by introducing MLP layers in the middle layer inputs of the decoder LLM.

**Strengths:**

- Well-written paper with excellent insights on the very timely topic of token representation learning for Vision-Language Models.
- The proposed multi-modal integration framework is relatively parameter-efficient as it shares the parameters of the self-attention layer with the additionally introduced cross-attention mechanism.

**Weaknesses:**

- While the first two conclusions in Sec. 3.3 seem convincing based on the analyses provided in Sec. 3.1-3.2, I am unsure how did the authors arrive at the third conclusion stating that the retention of Feed-Forward computations for fine-grained visual features in the
middle layers of the MLLM could yield significant benefits. Are specific experimental results or analyses that support this claim?
- Overall, the writing of the paper is good but can be improved by connecting the dots within the analyses. For instance, Sec. 3.2 analyses of "sparse and consistent attention .." can link the insights using entropy back to Fig. 2. This could give a consistent flow to the reader to understand the relation between the attention pattern/entropy, and the change in visual token representation.
- My major concern is a lack of ablation on the resource efficiency of the proposed integration framework. What are the additional memory/computational overhead in terms of FLOPs/wall-clock time and the required GPU memory? It seems that the additional MLP connections might also increase the number of parameters of the LLM significantly. Can the authors comment on this?

**Questions:**

- Why is it that the textual tokens exhibit constantly high yet relatively smaller cosine similarity per layer when compared to visual tokens (Fig. 2)? Can the authors present an intuitive explanation/insight into what leads to this phenomenon?
- Line 209: ".. visual tokens are selectively passed through either the Self-Attention ... ". Does the selectively passed visual tokens here refer to the visual anchors in Fig. 5? Can the authors rephrase this line to be more specific?
- For Sec. 3.2 analyses on visual-text interaction, it would be interesting to see what comprises the top-k visual/text tokens. Are these top-k tokens task-specific in particular, i.e., pretty much similar for different inputs of the same task as in [1] ? If so, then visual anchors might be an overkill. If no, then why?
- What concerns me is the how general the presented insights for visual tokens are for the textual modality? For example, can similar textual anchors be introduced for the text tokens as well? Will the decoding process benefit from this?

Overall, the paper is impressive in terms of the insights it presents. I would be willing to increase my score if the authors were to address the above questions/weaknesses.

Reference:

[1] Luo, Grace *et al.* “Task Vectors are Cross-Modal.” (2024).

---

### Official Review · Reviewer_bAN3 · 2024-11-02

**Soundness:** 2
**Presentation:** 3
**Contribution:** 3
**Rating:** 6
**Confidence:** 3

**Summary:**

This paper introduces a novel MLLM architecture named eRAM-V, designed to enhance the attention mechanism by utilizing a varying number of visual features across different layers of the MLLM. Through comprehensive analysis, it is observed that in both shallow and deep layers, the processing of visual tokens is often redundant. As a solution, this paper proposes selecting only a subset of visual tokens for the shallow layers, thereby reducing the overall computational load.

**Strengths:**

1. The analysis of interactions between visual and text tokens across various layers of MLLMs is well-motivated and provides valuable insights.
2. The proposed eRAM-V architecture is clearly illustrated and easy to understand.
3. The experiments denmonstrate improvements across various datasets while maintaining reduced computational costs.

**Weaknesses:**

1. The conclusion that the transformation of visual tokens within LLMs is largely redundant in both shallow and deep layers needs to be validated on a broader set of datasets.
2. The method for determining the number of visual anchor tokens isn’t clearly explained. It remains unclear if this number is dependent on specific tasks or datasets.

**Questions:**

see weakness

---

> ### Author Response · Authors · 2024-11-13
> **Rebuttal to reviewer bAN3**
>
> We sincerely thank you for your valuable feedback and constructive suggestions. We will address your concern below.
>
> **Q1:** The conclusion that the transformation of visual tokens within LLMs is largely redundant in both shallow and deep layers needs to be validated on a broader set of datasets.
>
> **R1:** Thanks for your suggestion. We have added visualization for more dataset in **Appendix J**.
>
> **Q2:** The method for determining the number of visual anchor tokens isn't clearly explained. It remains unclear if this number is dependent on specific tasks or datasets.
>
> **R2:** The number of visual anchors are fixed in our model. In this work, we select 65 visual anchors based on the weighted attention of the [CLS] token in the Vision Transformer (ViT). Specifically, we rank the visual tokens by their attention to the [CLS] token, where attention is averaged across the attention head dimension. (from b×h×1×n to b×1×n). The top 64 visual tokens, along with the [CLS] token, are then selected to form the final set of 65 visual anchors.
>
> We evaluate three configurations in our experiments: 65, 145, and 257 visual anchors. With fine-grained interaction through cross-attention, these configurations result in similar performance. However, when fine-grained interaction is removed, performance consistently improves as the number of visual anchors increases from 65 to 257. Despite this, we choose the 65-anchor configuration, as it offers a more efficient solution while still benefiting from the cross-attention mechanism.
>
> Thanks agagin for you work! If there is still any question, please let us know!

---

> ### Comment · Reviewer_bAN3 · 2024-11-27
>
> I appreciate the authors’  detailed explaination, which partially addressed my concerns, however, based on the disscusion from other reviewers, I will keep my original rating.

---

### Meta-Review · Area_Chair_UwJg · 2024-12-20

**Metareview:**

This paper presents eRAM-V, a multimodal model designed to optimize vision-language fusion by selectively integrating visual features in key layers to enhance efficiency. Through an in-depth analysis, it identifies that fine-grained vision-text interactions are most effective in the middle layers of the network. To reduce computational overhead, eRAM-V proposes utilizing only a subset of visual tokens in the shallow layers, streamlining the overall processing while maintaining performance. The paper addresses a topic of large interest to the community: VLMs and efficiency within these models. The paper is also has experiments that do show the reduction in computational costs. However, the reviewers raised several points that remain weaknesses of the paper after the rebuttal: the paper's writing and organisation should be improved, the unclear importance of keeping only visual anchors while not doing a similar strategy for text, and the (even after discussion) unclarity and inconsistency of high attention weights but redundancy for layers 0-2. and comparisons to prior works FastV and VTW. The AC recognises the thorough work in discussing the positives and negatives and the work in rebutting, but sees it as just below the bar of acceptance and thus recommends rejection. The AC hopes the authors include the feedback and resubmit to another conference.

**Additional Comments On Reviewer Discussion:**

There was extensive discussions in particular between v1GT and the authors, in particular about questions relating to consistency and contradicting results from other papers. Despite multiple back and forth, the concerns raised above have remained.

---

### Decision · Program_Chairs · 2025-01-22

Reject